# SCAFFOLDING A STUDENT TO INSTILL KNOWLEDGE

**Anil Kag**[*]
anilkag@bu.edu

**Durmus Alp Emre Acar**[*]
alpacar@bu.edu

**Aditya Gangrade**[†]
agangra2@andrew.cmu.edu

**Venkatesh Saligrama**[*]
srv@bu.edu

## ABSTRACT

We propose a novel knowledge distillation (KD) method to selectively instill teacher knowledge into a student model motivated by situations where the student's capacity is significantly smaller than that of the teachers. In vanilla KD, the teacher primarily sets a predictive target for the student to follow, and we posit that this target is overly optimistic due to the student's lack of capacity. We develop a novel scaffolding scheme where the teacher, in addition to setting a predictive target, also scaffolds the student's prediction by censoring hard-to-learn examples. The student model utilizes the same information as the teacher's soft-max predictions as inputs, and in this sense, our proposal can be viewed as a natural variant of vanilla KD. We show on synthetic examples that censoring hard-examples leads to smoothening the student's loss landscape so that the student encounters fewer local minima. As a result, it has good generalization properties. Against vanilla KD, we achieve improved performance and are comparable to more intrusive techniques that leverage feature matching on benchmark datasets.

## 1 INTRODUCTION

A fundamental problem in machine learning is to design efficient and compact models with near state-of-the-art (SOTA) performance. Knowledge Distillation (KD) (Zeng & Martinez, 2000; Bucila et al., 2006; Ba & Caruana, 2014; Hinton et al., 2015) is a widely used strategy for solving this problem wherein the knowledge from a large pre-trained teacher model with SOTA performance is distilled onto a small student network.

**Vanilla KD.** Hinton et al. (2015) proposed the popular variant of KD by matching the student soft predictions, $\mathbf{s}(\mathbf{x})$ with that of the pre-trained teacher, $\mathbf{t}(\mathbf{x})$ on inputs $\mathbf{x}$. Informally, during student training, an additional loss term, $D_{KL}(\mathbf{t}(\mathbf{x}), \mathbf{s}(\mathbf{x}))$ is introduced that penalizes the difference between student and teacher predictive distributions using Kullback-Leibler (KL) divergence. This promotes inter-class knowledge learned by the teacher. We will henceforth refer to Vanilla KD as KD.

**Capacity mismatch between student and teacher.** One of the primary issues in KD is that the loss function is somewhat blind to the student's capacity to interpolate. In particular, when the student's capacity is significantly lower than the teacher's, we expect the student to follow the teacher only on those inputs realizable by the student.

We are led to the following question: *What can the teacher provide by way of predictive hints for each input, so that the student can leverage this information to learn to its full capacity?*

**Our Proposal: Scaffolding a Student to Distill Knowledge (DiSK).** To address this question, we propose that the teacher, during training, not only set a predictive target, $\mathbf{t}(\mathbf{x})$, but also provide hints on hard to learn inputs. Specifically, the teacher utilizes its model to output a guide function, $g(\mathbf{x})$, such that the student can selectively focus only on those examples that it can learn.

- *if $g(\mathbf{x}) \approx 1$, teacher discounts loss incurred by the student on the input $\mathbf{x}$.*
- *if $g(\mathbf{x}) \approx 0$, teacher signals the input $\mathbf{x}$ as learnable by student.*

---

[*]ECE Department, Boston University, Boston, MA
[†]Statistics Department, Carnegie Mellon University, Pittsburgh, PA

With this in mind we modify the KL distance in the KD objective and consider, $D_{KL}(\mathbf{t}(\mathbf{x}), \phi(\mathbf{s}(\mathbf{x}), g(\mathbf{x})))$, where $\phi(\mathbf{s}, g)$, which will be defined later, is such that, $\phi(\mathbf{s}, 0) = \mathbf{s}$ if $g$ offers no scaffolding. We must impose constraints on the guide function $g$ to ensure that only hard-to-learn examples are scaffolded. In the absence of such constraints, the guide can declare all examples to be hard, and the student would no longer learn. We propose to do so by means of a budget constraint $B(\mathbf{s}, g) \leq \delta$ to ensure that the guide can only help on a small fraction of examples. While more details are described in Sec. 3, we note that, in summary, our proposed problem is to take the empirical linear combination of the aforementioned KL distance and a cross-entropy term as the objective, and minimize it under the empirical budget constraint.

We emphasize that $g(\mathbf{x})$ is used only during training . The inference logic for the student remains the same as there is no need for $g(\mathbf{x})$ during inference. The guide function supported student training has three principal benefits. The benefits are explored in Sec. 2.

• **Censoring Mechanism.** Our guide function censors examples that are hard to learn for the student. In particular, when there is a large capacity gap, it is obvious that the student cannot fully follow the teacher. For this reason, the teacher must not only *set an expectation* for the student to predict, but also selectively gather examples that the student has the *ability to predict.*
• **Smoothen the Loss landscape.** We also notice in our synthetic experiments that whenever scaffolding is powerful, and can correct student's mistakes, the loss landscape undergoes a dramatic transformation. In particular, we notice fewer local minima in the loss viewed by the guided student.
• **Good Generalization.** The solution to our constrained optimization problem in cases where the guide function is powerful can ensure good student generalization. Specifically, we can bound the statistical error in terms of student complexity and not suffer additional complexity due to the teacher.

**Contributions.** We summarize our main results.

• We develop a novel approach to KD that exploits teacher representations to adjust the predictive target of the student by scaffolding hard-to-learn points. This novel scaffolding principle has wider applicability across other KD variants, and is of independent interest.
• We design a novel response-matching KD method (Gou et al., 2021) which is particularly relevant in the challenging regime of large student-teacher capacity mismatch. We propose an efficient constrained optimization approach that produces powerful training scaffolds to learn guide functions.
• Using synthetic experiments, we explicitly illustrate the structural benefits of scaffolding. In particular, we show that under our approach, guides learn to censor difficult input points, thus smoothening the student's loss-landscape and often eliminating suboptimal local minima in it.
• Through extensive empirical evaluation, we demonstrate that the proposed DiSK method;
– yields large and consistent accuracy gains over vanilla KD under large student-teacher capacity mismatch (upto 5% and 2% on CIFAR-100 and Tiny-Imagenet).
– produces student models that can get near-teacher accuracy with significantly smaller model complexity (e.g. $8\times$ computation reduction with $\sim 2\%$ accuracy loss on CIFAR-100).
– improves upon KD even under small student-teacher capacity mismatch, and is even competitive with modern feature matching approaches.

## 2 ILLUSTRATIVE EXAMPLES

We present two synthetic examples to illustrate the structural phenomena of the censoring mechanism and smoothening of student's loss landscape enabled by the scaffolding approach DiSK, which lead to globally optimal test errors. We defer exact specification of the algorithm to Sec.3.

**Example 1 (1D Intervals).** Consider a toy dataset with one dimensional features $x \in [0, 9]$ and binary class labels $y \in \{\text{Red}, \text{Blue}\}$ as shown in Figure 1. There are two Blue labelled clusters as in $[2, 3]$ and in $[5, 7]$. The remaining points are labelled as Red. We sample 1000 i.i.d. data points as the training set and 100 data points as the test set with balanced data from both classes. We describe the details of the experiment setup such as models and learning procedure in Appx. A.1.

Teacher $T$ belongs to the 2-interval function class, and the capacity-constrained student $S$ belongs to the 1-interval function class. Since teacher capacity is sufficient to separate the two classes without error, it learns the correct classifier (see Figure 1). In contrast, the best possible student hypothesis cannot correctly separate the two classes. Hence, the student will have to settle onto one of the many

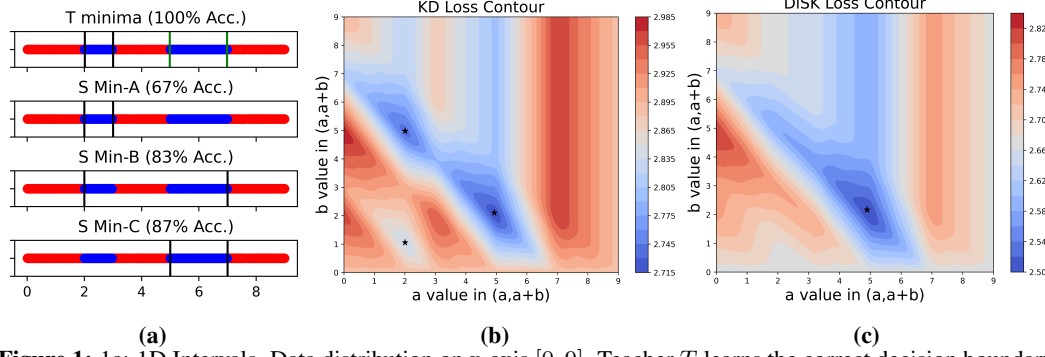

**(a)** **(b)** **(c)**

**Figure 1:** 1a: 1D Intervals. Data distribution on x-axis $[0, 9]$. Teacher $T$ learns the correct decision boundary with 2-intervals and it is the global minima for this binary classification task. Student $S$ has many bad local minima, and one global minima that best describes the decision boundary with 1-interval. 1b: KD training. Loss contour plot shows the various local minima exist. 1c: DiSK training. Loss contour plot shows the bad local minima no longer exist.

**Table 1:** The number of times each method lands on various local minima in two toy problems for 100 runs.

| Dataset | 1D Intervals | | | 2D Gaussians | | | |
|---|---|---|---|---|---|---|---|
| **Minima** | **A** | **B** | **C (Global)** | **A** | **B** | **C** | **D (Global)** |
| **Accuracy** | **67%** | **83%** | **87%** | **70%** | **80%** | **90%** | **100%** |
| Cross-Entropy | 35 | 64 | 1 | 73 | 12 | 9 | 6 |
| KD | 30 | 67 | 3 | 1 | 11 | 31 | 57 |
| DiSK | 9 | 1 | 90 | 0 | 0 | 3 | 97 |

local minima. We show these minima and the contour plot for the student in Figure 1. We present the results of training student models with different initializations in Table 1.

*KD suffers from bad local minima.* KD loss landscape contains many local minima (see Figure 1). Due to a big gap between student and teacher capacity, it is unable to help the student discern between these minima. Hence, KD fails to distinguish between the different minima (see Table 1).

*DiSK censors interval $[2, 3]$ and in addition focuses training on learnable datapoints.* If we analyze the guide function at the end of the training, we see that it covers (censors) the first Blue cluster. Indeed, both clusters are not simultaneously *l*earnable with the available student capacity. Once we censor the interval $[2, 3]$, then the problem becomes realizable for the student model. The guide function thus captures the excess capacity of the data.

*DiSK smoothens loss landscape.* The guide function and the budget constraint enable our method to have a smooth loss landscape thanks to the guide-function censoring points, which eliminate the local minima. Hence, DiSK solution lands in the global minimum with high probability.

**Example 2 (2D Gaussians).** Consider another toy dataset with two dimensional features $\mathbf{x} \in \mathbb{R}^2$ and three class labels $y \in \{\text{Red}, \text{Green}, \text{Blue}\}$. Here we wish to show that DiSK can allow for *globally optimal solutions reaching 100% accuracy, which appears unachievable with cross-entropy minimization regardless of data size.*

Figure 2a shows the labelled data. There are six cluster centers, two with each class label. Data points are drawn using Gaussian balls around the cluster centers with small radii. We sample $1000$ i.i.d. data points as the training set and $1000$ data points as the test set with equal representation from all three classes. We provide details (hypothesis classes, learning procedure, etc.) in Appx. A.2.

The teacher is a 3-layer neural network with 8, 16, and 3 neurons. The student is a 2-layer neural network with 2 and 3 neurons. We point out that the teacher being an over-parameterized network in this feature space, easily learns the correct decision boundary. While the student being severely constrained network suffers in learning the task. Different training runs lead to different local minima. We show teacher solution and student local minima in Figure 2a. The contour plots for the student models under KD loss and DiSK loss are shown in Figure 2b-2c using Li et al. (2018).

The results are similar to the 1D example - KD converges to a poor local minimum with at least 43% of the initializations, while in contrast, DiSK escapes these by focusing on the learnable part of the input space (Fig. 2c), converging to the global minimum nearly always (Table 1).

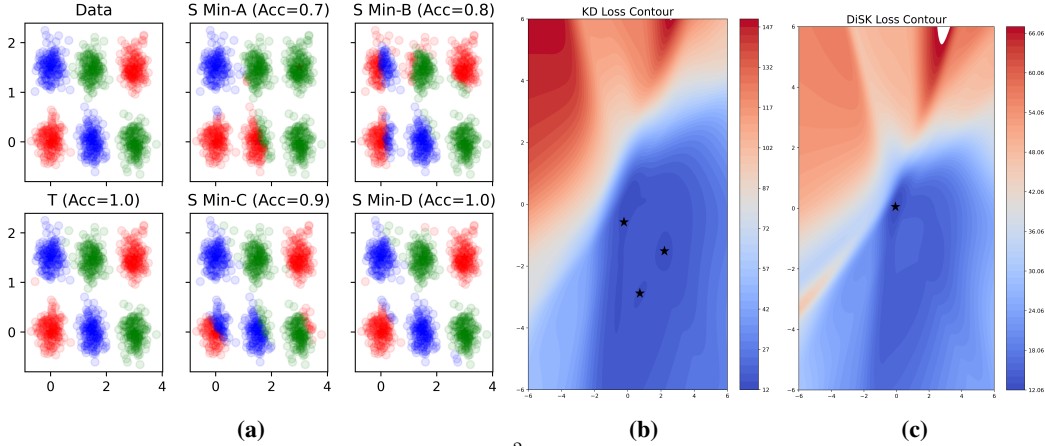

**(a)**                                   **(b)**                            **(c)**

**Figure 2:** 2a. 2D Gaussians. Data distribution on $\mathbb{R}^2$. Teacher $T$ learns the correct decision boundary with 3 layer NN and it is the global minima for this three-way classification task. While student $S$ has many bad local minima, and one global minima that best describes the decision boundary with 2 layer NN. 2b. KD training. Loss contour plot shows the various local minima in the loss landscape. 2c. DiSK training. Loss contour plot shows the bad local minima no longer exist (wider minima, join two adjust minima, remove bad local minima).

To conceptualize our findings in these examples, let us attempt to intuitively infer the example-censoring, landscape-smoothening, and good generalization, by utilizing the following conditions that appear to be satisfied for these synthetic examples.

*Realizability.* Suppose we are in a situation where the guide function $g \in \mathcal{G}$ is sufficiently powerful that there is a student and guide function capable of interpolation, i.e., predictions supported by the guide function, $\phi(\mathbf{s}, g)$, interpolates to mimic the labels.
*Example:* For instance, consider a binary classification problem with the labels $y \in \{-1, 1\}$. Let $\phi(s, g) = y(s + g)$ with $s(\mathbf{x}) \in [-1, 1]$. Our realizability condition is that we always satisfy $y(s(\mathbf{x}) + g(\mathbf{x})) > 0$. As such, this leads to the condition that if $ys(\mathbf{x}) \leq 0$, then $yg(\mathbf{x}) > 0$. Therefore, $\mathbb{E}[\mathbb{1}_{[ys(\mathbf{x}) < 0]}] \leq \mathbb{E}[\mathbb{1}_{[ys(\mathbf{x}) < 0, yg(\mathbf{x}) > 0]}] \leq \mathbb{E}[\mathbb{1}_{[yg(\mathbf{x}) > 0]}]$.

*Small Guide Function Capacity.* In addition to realizability suppose the class of guide functions $g \in G$ has a small capacity (for instance, small VC dimension). For our case this condition is satisfied because our guidance function is obtained by using an MLP on teacher's last layer features.
*Example:* Continuing with the example above, say we now have $m$ training instances, $(\mathbf{x}_i, y_i)$, $i \in [m]$, $\hat{g}(\mathbf{x})$ is guide function output of DiSK. We can infer by standard statistical learning results (Shalev-Shwartz & Ben-David, 2014) that, for the estimated function $\hat{g} \in G$, it follows with probability greater than $1 - \eta$ that $\mathbb{E}[\mathbb{1}_{[y\hat{g}(\mathbf{x}) > 0]}] \leq \frac{1}{m} \sum_{i=1}^{m} \mathbb{1}_{[y_i \hat{g}(\mathbf{x}_i) > 0]} + \sqrt{\frac{VC(G) + \log \frac{1}{\eta}}{m}}$. As a result, we can say that if there is a student, $s(\mathbf{x})$ (not necessarily that output by DiSK), which complements $\hat{g}(\mathbf{x})$ and satisfies realizability, then with probability greater than $1 - \eta$:
$$\mathbb{E}[\mathbb{1}_{[ys(\mathbf{x}) < 0]}] \leq \frac{1}{m} \sum_{i=1}^{m} \mathbb{1}_{[y_i \hat{g}(\mathbf{x}_i) > 0]} + \mathcal{O}\left(\sqrt{\frac{VC(G) + \log \frac{1}{\eta}}{m}}\right).$$

*Remarks.* The key point is that the student capacity is considerably larger since we typically train an entire DNN, and student complexity-based bound can be vacuous. While the guidance function does bound the student generalization error in terms of guide function complexity, there are strong caveats— we require the strong assumption of realizability on the entire domain, and additionally, while the guide function can witness student error, we are not in a position to precisely estimate it without additional training data. Furthermore, the RHS is a relaxed bound on the student training error. This motivates having a budget constraint to ensure that student learns with small training error.

## 3 Definitions and Formulations

*Notation.* Let $\mathcal{X}$ and $\mathcal{Y} = \{1, \ldots, C\}$ be the feature and label spaces respectively, focusing on a $C$-class classification task. We assume that we have a training set of $N$ i.i.d. data points $\mathcal{D} = \{\mathbf{x}_i, y_i\}_{i=1}^{N}$, where $\mathbf{x}_i \in \mathcal{X}$ and $y_i \in \mathcal{Y}$. We use symbols $S$ and $T$ to denote the student and teacher models respectively. Let $\mathbf{l}_S(\mathbf{x}) \in \mathcal{R}^{|\mathcal{Y}|}$ and $\mathbf{l}_T(\mathbf{x}) \in \mathcal{R}^{|\mathcal{Y}|}$ be the score vector, logits, predicted by $S$ and $T$

on input $\mathbf{x}$. We use $\tau$ as the temperature used to soften the probability distribution. We write the resulting softened student and teacher probabilities as $\mathbf{s}^\tau(\mathbf{x})$ and $\mathbf{t}^\tau(\mathbf{x})$, i.e.,

$$\mathbf{s}^\tau(\mathbf{x}) = \mathrm{softmax}\left(\frac{\mathbf{l}_S(\mathbf{x})}{\tau}\right); \quad \mathbf{t}^\tau(\mathbf{x}) = \mathrm{softmax}\left(\frac{\mathbf{l}_T(\mathbf{x})}{\tau}\right)$$

The standard prediction probabilities correspond to $\mathbf{s}^1(\mathbf{x})$ and $\mathbf{t}^1(\mathbf{x})$. We will use $\mathbf{s}_y^\tau(\mathbf{x})$ to denote the $y$th coordinate in $\mathbf{s}^\tau(\mathbf{x})$, and similarly $\mathbf{t}_y^\tau(\mathbf{x})$. The hard prediction of the student is $p_S(\mathbf{x}) = \arg\max_{y \in \mathcal{Y}} \mathbf{s}_y^1(\mathbf{x})$, and similarly $p_T(\mathbf{x}) = \arg\max_{y \in \mathcal{Y}} \mathbf{t}_y^1(\mathbf{x})$ for the teacher.

We use $g(\mathbf{x}) \in [0, 1]$ to denote the helper guide function for the student and teacher pair $(S, T)$. Guide takes input $\mathbf{x}$ and any other feature processed by $(S, T)$ pair and decides whether or not the student needs help on the input $\mathbf{x}$. Finally, we define ReLU activation as $(\cdot)_+ = \max(0, \cdot)$.

## 3.1 Vanilla Knowledge Distillation

KD relaxes the $0-1$ error between the student predictions and the true labels $y$ using the cross-entropy loss $\mathcal{L}_{CE}$. Then, KD denotes the distance between the student and teacher softened probability distributions using the KL divergence. We summarize the corresponding losses as,

$$\mathcal{L}_{CE}(\mathbf{s}) = -\frac{1}{N}\sum_{i=1}^{N} \log \mathbf{s}_{y_i}^1(\mathbf{x}_i); \quad \mathcal{L}_{KL}^\tau(\mathbf{s}) = -\frac{1}{N}\tau^2\sum_{i=1}^{N}\sum_{y} \mathbf{t}_y^\tau(\mathbf{x}_i) \log \frac{\mathbf{s}_y^\tau(\mathbf{x}_i)}{\mathbf{t}_y^\tau(\mathbf{x}_i)}$$

For hyperparameters $\alpha \in [0, 1], \tau > 0$, KD minimizes a mixture of the above losses, as shown below

$$\mathcal{L}_{KD}^{\tau,\alpha}(\mathbf{s}) = \alpha\mathcal{L}_{CE}(\mathbf{s}) + (1-\alpha)\mathcal{L}_{KL}^\tau(\mathbf{s}). \tag{1}$$

## 3.2 Selective Knowledge Distillation.

KD attempts to transfer the knowledge from the teacher to the student on all training data points, which is a sub-optimal objective when there is a capacity mismatch between the student and the teacher. Instead, we propose distilling selective knowledge (DiSK) to allow the student to selectively ignore some hard-to-learn data points during training, transferring the teacher's knowledge only on easy-to-learn inputs, and matching the learning to student capacity. Our objective is to minimize

$$\min_{\mathbf{s},g,\delta} \underbrace{\frac{1}{N}\sum_{i=1}^{N}\mathrm{distance}(\mathbf{t}(\mathbf{x}_i); \phi(\mathbf{s}, g)(\mathbf{x}_i))}_{\text{Distance between } T \text{ and } S \text{ with help of } g} \quad \text{subject to.} \quad \underbrace{\frac{1}{N}\sum_{i=1}^{N}g(\mathbf{x}_i)\mathbb{1}_{\{y_i \neq \arg\max_y \mathbf{s}_y(\mathbf{x}_i)\}} \leq \delta}_{\text{Support budget constraint on } g} \tag{2}$$

where $\phi$ interpolates student predictions based on the guide's help. The divergence term helps in minimizing the distributional distance between the teacher and student probabilities after the guide $g$ is included. While the budget term in the optimization constrains the helper $g$ to provide help only when necessary, the amount of help given to the student should be within the budget $\delta \in [0, 1]$.

*Function $g$ Construction.* As previously stated, we use the teacher's last layer features and soft predictions as input to the guide $g$. The guide is structured as a light-weight three-layer neural network with these inputs, with a sigmoid activation at the last layer. We re-emphasise that $g$ is not used at inference time, and only aids training. More details are left to Appx.A.4.

**Relaxed Losses, Lagrangian & Optimization Algorithm.** We relax Eq. 2 and construct a Lagrangian by integrating the constraint into the minimization.

*Budget constraint relaxation.* We relax the indicator loss in the budget to a cross-entropy, and treat $\delta$ as a hyperparameter to get

$$\mathcal{L}_{budget}^\delta(\mathbf{s}, g) = \left[-\frac{1}{N}\sum_{i=1}^{N}g(\mathbf{x}_i)\log \mathbf{s}_{y_i}^1(\mathbf{x}_i) - \delta\right]_+ \tag{3}$$

We view the scaffold as a way for the student to interpolate the uncensored data. It suggests that a good initialization for the budget is the error of cross-entropy trained model when the student does not have the teacher supervision. Thus, we scan the budget in a small interval around this initialization.

---

**Algorithm 1** DiSK: Distilling Selective Knowledge.

---

1: **Input:** Training data $\mathcal{D} = \{(\mathbf{x}_i, y_i)\}_{i=1}^N$, Teacher $\mathbf{t}$,
2: **Parameters:** $\tau$, $K$, $\alpha$, $\lambda_{\min}$, $\lambda_{\max}$, Number of iterations $R$, $\lambda_T$ cosine period, Budget $\delta$,
3: **Initialize:** $\mathbf{s}$, randomly initialize $g$, $\lambda = \lambda_{\min}$,
4: **for** $r = 1$ **to** $R$ **do**
5:    Randomly Shuffle Dataset $\mathcal{D}$
6:    $g \leftarrow \arg\min_g \mathcal{L}_{DiSK}^{\tau,K,\delta,\alpha}(\mathbf{s}, g, \lambda)$
7:    $\mathbf{s} \leftarrow \arg\min_{\mathbf{s}} \alpha \mathcal{L}_{CE}(\mathbf{s}) + (1-\alpha)\mathcal{L}_{dist}^{\tau,K}(\mathbf{s}, g)$
8:    $\lambda \leftarrow \lambda_{\min} + (\lambda_{\max} - \lambda_{\min}) \times \frac{(1-\cos\frac{r \bmod \lambda_T}{\lambda_T}\pi)}{2}$
9: **Return : s**

---

*Distillation objective.* Motivated from KL loss, we construct a distance loss with guide function as,

$$\mathcal{L}_{dist}^{\tau,K}(\mathbf{s}, g) = -\frac{1}{N}\tau\tau_{\mathbf{t},\mathbf{s},\mathcal{D}} \sum_{i=1}^N \sum_y t_y^\tau(\mathbf{x}_i) \log\left(s_y^{\tau_{\mathbf{t},\mathbf{s},\mathcal{D}}}(\mathbf{x}_i) + 1_{y \in \text{top}_K(t^\tau(\mathbf{x}_i))} g(\mathbf{x}_i)\right) \tag{4}$$

We point out two modifications in the distillation loss. First, $\mathcal{L}_{dist}$ explicitly adds guide value to softened student probabilities in selected class indices. The class indices guide function adds value are picked as top $K$ classes based on the teacher probabilities for any input $\mathbf{x}_i$ where $K$ is a hyperparameter of our method. The rest of the class indices do not get any value from $g$. Second, we use different temperature parameters for teacher and student. Temperature parameter for teacher, $\tau$, is a hyperparameter. The student temperature is found by minimizing the KL loss between teacher softened probabilities and the student softened probabilities over the training dataset, .i.e $\tau_{\mathbf{t},\mathbf{s},\mathcal{D}} = \arg\min_{\tau'} \sum_i KL(\mathbf{t}^\tau(\mathbf{x}_i), \mathbf{s}^{\tau'}(\mathbf{x}_i))$.

Similar to KD, we incorporate standard cross entropy loss between student model predictions and the ground truth labels for stability. We construct our Lagrangian by combining Eq. 3 and 4 as,

$$\mathcal{L}_{DiSK}^{\tau,K,\delta,\alpha}(\mathbf{s}, g, \lambda) = \alpha\mathcal{L}_{CE}(\mathbf{s}) + (1-\alpha)\mathcal{L}_{dist}^{\tau,K}(\mathbf{s}, g) + \lambda\mathcal{L}_{budget}^\delta(\mathbf{s}, g) \tag{5}$$

where $\alpha$ is a hyper-parameter and $\lambda$ is the dual parameter of DiSK.

We optimize Obj. 5 using a primal dual update scheme as explained in Algorithm 1.

*Primal Parameter Updates (*$\mathbf{s}$, $g$*).* We learn the student $\mathbf{s}$ and the guide function $g$ using alternating minimization. We approximate $\arg\min$ with running SGD for a small number of epochs on the full dataset. In each iteration, we first learn the guide function $g$ to select the data partition from which the knowledge needs to be distilled. Next, given the function $g$, we learn the student using the help $g$. We empirically found that not optimizing the student model on budget loss gives more stable results. Hence, we minimize the student model only on the distillation and cross-entropy losses.

*Dual Parameter Update (*$\lambda$*) Intuition.* Although it is tempting to optimize the above via a dual ascent and primal descent scheme (wherein the dual parameter $\lambda$ is increased by residual term in the constraint until constraint satisfaction), recent work, Sun & Sun (2021), has proposed to decrease the $\lambda$ in the non-convex regime. Inspired by this, we update the dual parameter by a fixed schedule between $[\lambda_{\min}, \lambda_{\max}]$. $\lambda_{\min}$ encourages exploration and allows student model to distill knowledge from all points. On the other hand, $\lambda_{\max}$ enforces the constraint and forces the student model to learn on uncensored inputs. We choose $R \approx 4\lambda_T$, so that the algorithm is exposed to a few exploratory periods. For the final period, we increase $\lambda$ monotonically so that budget is more strictly enforced at termination.

*Computational Efficiency.* Algorithm 1 trains both student and guide networks. The guide network being small (three-layer MLP) relative to the student (CNN model), the additional cost in training the guide is relatively insignificant, and as such DiSK efficiency is similar to KD.

## 4 EXPERIMENTS

We evaluate DiSK in various capacity mismatch scenarios on benchmark datasets. We avail our code at `https://github.com/anilkagak2/DiSK_Distilling_Scaffolded_Knowledge`

**Datasets.** We use publicly available CIFAR-100 (Krizhevsky, 2009), Tiny-Imagenet (Le & Yang, 2015) datasets. CIFAR-100 contains 50K training and 10K test images from 100 classes with size

**Table 2:** Model Statistics. We compute the storage (number of parameters) and computational requirements (number of multiply-addition operations) of the models used in this work.

| Architecture | | CIFAR-100 | | Tiny-Imagenet | | Architecture | CIFAR-100 | |
|---|---|---|---|---|---|---|---|---|
| | | MACs | Params | MACs | Params | | MACs | Params |
| **Teacher** | ResNet10-$\ell$ | 64M | 1.25M | 255M | 1.28M | | | |
| | ResNet10 | 253M | 4.92M | 1013M | 5M | ResNet32x4 | 1083M | 7.4M |
| | ResNet18 | 555M | 11.22M | 2221M | 11.27M | Wide-ResNet-40-2 | 327M | 2.25M |
| | ResNet34 | 1159M | 21.32M | 4637M | 21.38M | | | |
| **Student** | ResNet10-xxs | 2M | 13K | 8M | 15K | ResNet8x4 | 177M | 1.2M |
| | ResNet10-xs | 3M | 28K | 12M | 31K | ShuffleNetV2 | 44.5M | 1.4M |
| | ResNet10-s | 4M | 84K | 16M | 90K | Wide-ResNet-16-2 | 101M | 700K |
| | ResNet10-m | 16M | 320K | 64M | 333K | Wide-ResNet-40-1 | 83M | 570K |
| | | | | | | MobileNetV2x2 | 22M | 2.4M |

$32 \times 32 \times 3$. While Tiny-Imagenet contains 100K training and 10K test images from 200 classes with size $64 \times 64 \times 3$. We provide the dataset setup and data augmentations used in detail in Appx. A.3.

**Models.** We evaluate standard convolutional models on these datasets including ResNet(He et al., 2016), Wide-ResNet(Zagoruyko & Komodakis, 2016), MobileNet(Sandler et al., 2018), and Shuffle-Net(Ma et al., 2018). Table 2 shows the storage and computational requirements of all the models used in this work. We provide explicit model configurations in Appx. A.4, including the tiny models we generate from the ResNet architectures.

**Methods.** We study performance against standard cross-entropy (CE) based learning and the vanilla KD methods. For each method, we train models for 200 epochs using SGD as the optimizer with 0.9 momentum and 0.1 learning rate. See Appx. A.5 for more training details. We have recorded the mean in our results as the variance of 3 trials in our experiments is not larger than 0.1 in most cases.

We perform evaluations in different settings. Below, we explain individual setups. We cover more ablative experiments in Appx. A due to page limit.

**Large Capacity Mismatch Setting.** We distill knowledge from a teacher model into a student model where the student has much less capacity compared to the teacher model. We use four large capacity ResNet teachers and five tiny ResNet students and train these students using CE, KD, and DiSK methods. Performances, and the gains of DiSK are reported in Table 3.

**Small Capacity Mismatch Setting.** While DiSK has been designed for the scenario when student capacity is very low, we further evaluate it in the setting where teacher and student capacities are similar, to probe how far the power of the method extends. The model classes used are the standard choice for this scenario (Chen et al., 2022; Tung & Mori, 2019). Performance is reported in Table 4.

Table 4 further reports the results of the feature matching distillation methods: FitNets (Romero et al., 2015), SemCKD (Chen et al., 2021), and SimKD (Chen et al., 2022). Such methods can often outperform response matching KD on large students, due to student representations that are more aligned with the teacher, but typically at an increased training cost. While feature matching methods are not the main focus of our work (and in principle scaffolding idea can be extended to them), we observe that DiSK often improves upon their performance without any direct feature matching.

**Experiment results**. Below, we highlight salient features of DiSK based on empirical data.

*DiSK outperforms the baselines uniformly across all datasets and student sizes.* As shown by Table 3, DiSK significantly improves the student performance in CIFAR-100 and the (more challenging) Tiny-Imagenet dataset, respectively showing accuracy gains of up to 5% and 2% compared to KD. These gains are consistent across a wide range of student and teacher capacities.

*DiSK achieves better performance with worse teachers than KD does with even the best teachers.* In Table 3, we point out that the student performance increases for KD as the teacher complexity is increased for a given student. But note that for the same student, DiSK achieves much better performance with even worse teacher. For instance, for the 'ResNet10-m' student, KD accuracy increases from 66.96% to 68.09% by using high capacity teachers. But 'ResNet10-m' trained with even the worst teacher ( 'ResNet10-$\ell$' ) achieves 70.03% accuracy. This saves a lot of resources in any application as large teacher requires more training time, and larger compute resources.

*DiSK is competitive even in small capacity difference setting.* As shown by Table 4, DiSK does not loose its competitive edge over the KD even when the student is relatively similar sized as

**Table 3:** DiSK performance under large capacity mismatch on CIFAR-100 & Tiny-Imagenet: We draw mismatched teachers and students from the ResNet family, and report accuracy of CE trained teachers and students, performance of students distilled using KD and DiSK, and gains of the latter relative to KD.

| Architecture | | CIFAR-100 | | | | | Tiny-Imagenet | | | | |
| | | Accuracy (%) | | | | | Accuracy (%) | | | | |
| Teacher | Student | Teacher | CE | KD | DiSK | Gain | Teacher | CE | KD | DiSK | Gain |
|---|---|---|---|---|---|---|---|---|---|---|---|
| ResNet10-$\ell$ | ResNet10-xxs | 71.99 | 32.05 | 32.64 | **37.56** | **4.92** | 52.14 | 17.44 | 17.59 | **18.62** | **1.03** |
| | ResNet10-s | | 52.16 | 54.92 | **58.14** | **3.22** | | 34.65 | 35.77 | **37.43** | **1.66** |
| | ResNet10-m | | 65.24 | 66.96 | **70.03** | **3.07** | | 44.74 | 46.01 | **48.03** | **2.02** |
| ResNet10 | ResNet10-xxs | 75.25 | 32.05 | 34.25 | **37.84** | **3.59** | 56.04 | 17.44 | 17.96 | **18.55** | **0.59** |
| | ResNet10-s | | 52.16 | 54.95 | **58.36** | **3.41** | | 34.65 | 36.11 | **37.37** | **1.26** |
| | ResNet10-m | | 65.24 | 67.27 | **70.15** | **2.88** | | 44.74 | 46.08 | **48.19** | **2.11** |
| ResNet18 | ResNet10-xxs | 76.56 | 32.05 | 34.16 | **37.8** | **3.64** | 62.48 | 17.44 | 17.47 | **18.53** | **1.06** |
| | ResNet10-s | | 52.16 | 55.76 | **58.11** | **2.35** | | 34.65 | 35.59 | **37.5** | **1.91** |
| | ResNet10-m | | 65.24 | 68.09 | **69.86** | **1.77** | | 44.74 | 45.91 | **47.7** | **1.79** |
| ResNet34 | ResNet10-xxs | 80.46 | 32.05 | 33.93 | **37.78** | **3.85** | 63.06 | 17.44 | 17.67 | **18.91** | **1.24** |
| | ResNet10-s | | 52.16 | 54.19 | **58.02** | **3.83** | | 34.65 | 35.43 | **37.68** | **2.25** |
| | ResNet10-m | | 65.24 | 66.78 | **69.89** | **3.11** | | 44.74 | 45.89 | **47.6** | **1.71** |

**Table 4:** DiSK performance with small capacity mismatch on CIFAR-100. We pick standard student and teacher configurations used in the KD literature, and report accuracies and gains similarly to Table 3. Feature matching KD baselines are due to Chen et al. (2022).

| Architecture | | CIFAR-100 | | | | | | | |
| | | Response Matching KD | | | | | Feature Matching KD | | |
| | | Accuracy (%) | | | | | Accuracy (%) | | |
| Teacher | Student | Teacher | CE | KD | DiSK | Gain | FitNet | SemCKD | SimKD* |
|---|---|---|---|---|---|---|---|---|---|
| ResNet32x4 | ResNet8x4 | 81.45 | 73.89 | 76.25 | **76.92** | **0.67** | 74.32 | 76.23 | 78.08 |
| | ShuffleNetV2 | | 73.74 | 79.13 | **80.23** | **1.1** | 75.82 | 77.62 | 78.39 |
| | Wide-ResNet-16-2 | | 74.26 | 76.28 | **77.67** | **1.39** | 74.70 | 75.65 | 77.17 |
| | MobileNetV2x2 | | 69.24 | 76.05 | **77.24** | **1.19** | 73.09 | 73.98 | 75.43 |
| Wide-ResNet-40-2 | ResNet8x4 | 78.41 | 73.89 | 75.15 | **76.05** | **0.9** | 75.02 | 75.85 | 76.75 |
| | ShuffleNetV2 | | 73.74 | 75.81 | **78.33** | **2.52** | - | - | - |
| | Wide-ResNet-40-1 | | 72.81 | 74.44 | **75.92** | **1.48** | 74.17 | 74.4 | 75.56 |
| | MobileNetV2x2 | | 69.24 | 73.92 | **76.32** | **2.40** | - | - | - |

*SimKD accuracy is not emphasized as it employs additional layers beyond the given student architecture and thus not directly comparable to other methods.

teachers, and shows gains of up to $2.5\%$ relative to KD. We conjecture that the observed gains arise from the fact that DiSK provides scaffolding for hard points to the student in initial training stages, which promotes the student to learn easy examples first. As training progresses, DiSK removes the discounted help from hard inputs. As a result, the student evolves from simpler hypothesis to the ones consistent with both easy and hard inputs. This justifies our dual parameter ($\lambda$) update in Algorithm 1, wherein we periodically increase and decrease $\lambda$ to enforce and relax the budget constraint.

*DiSK students achieve near teacher accuracy while saving up to $8\times$ MACs & $5\times$ Params.* As reported in Table 3, student ('ResNet10-m') trained with the teacher ('ResNet10-$\ell$') achieves close to the teacher accuracy of $71.99\%$. In this process, it saves $4\times$ compute and requires $4\times$ less parameters. Similarly, student ('ShuffleNetV2') trained with the teacher ('ResNet32x4') achieves close to the teacher accuracy of $81.45\%$. In this process, it saves $24\times$ compute and requires $5\times$ less parameters.

*DiSK cleverly selects a subset of datapoints and smoothens the loss landscape.* As illustrated in Figure 1 and 2, DiSK judiciously selects a subset of hard-to-learn data points for the students and provide discounted help to the student focus on easily learnable inputs. As a result, it eliminates some bad local minima in the student loss-landscape, and smoothens tihs surface.

*DiSK enables the student to reach saturation capacity.* In Table 3, the performance of KD often suffers as the teacher size is increased, e.g., student ('ResNet10-s') accuracy decreases substantially with the teacher 'ResNet34' versus the teacher 'ResNet18'. In contrast, DiSK saturates the student performance across different teachers. For instance, student ('ResNet10-m') accuracy is $\approx 70\%$ for all the teachers. Thus, we point out that DiSK enables the student to reach saturation. This may be due to the fact that guide $g$ identifies the same set of 'easy' points across different teachers.

## 5 RELATED WORK

We refer the reader to Gou et al. (2021) for a comprehensive survey on knowledge distillation.

**Response Matching.** Zeng & Martinez (2000); Bucila et al. (2006) distill the response of an ensemble of classifiers into a single neural network by creating a pseudo-labeled dataset using the ensemble of classifiers. Ba & Caruana (2014) extend this to the setting with the neural network as the teacher. Hinton et al. (2015) propose vanilla KD that distills knowledge from an ensemble of neural networks into a single network by matching their output logits. This work provided a simple recipe for aligning the teacher and student predictive distributions using the Kullback-Leibler (KL) divergence. Recently, Beyer et al. (2021) modify the KD procedure to include patient and consistent teacher resulting in substantial gains. Knowledge consistency is enforced by using the same aggressive data augmentation and image views in the student as in the teacher. Patience is promoted using a very long training schedule. This results in a computationally very expensive training process.

Stanton et al. (2021) analyze response matching KD and suggests that difficulty in optimization leads to poor knowledge distillation. Thus, the teacher and student predictions do not always match, even on the training data. Cho & Hariharan (2019) study vanilla KD through the lens of mismatched student and teacher capacities. They show that small students are unable to mimic complex teachers. They propose early stopping teacher training as to remedy to achieve a student-learnable teacher. The above works fail miserably when the gap between student and teacher complexities is large. Specifically, the student cannot learn the complex teacher decision boundaries primarily due to the small student capacity. It becomes imperative to selectively choose only easy-to-learn data points and transfer the teacher knowledge from these points and ignore the hard-to-learn data points during distillation. Thus, our proposal targets the problem of severe capacity gaps between student and teacher models. Additionally, our experiments with standard student and teacher configurations show that DiSK is still competitive when the capacity difference is small.

**Feature Matching.** In response matching, teacher supervision is limited to its logits. We can enforce intermediate layer feature matching for refined teacher supervision. FitNets (Romero et al., 2015) extend the KD by including the feature matching in the middle layers. Zagoruyko & Komodakis (2017) use feature map attention as the teacher supervision. Tung & Mori (2019) preserve pairwise similarity in feature maps amongst data points during the distillation. Chen et al. (2022) modify the student by projecting the student features onto the teacher feature space and by reusing the teacher classifier. While our work focuses primarily on selective distillation in vanilla KD for simplicity. We can easily extend the proposed framework to incorporate it into feature-matching distillation.

**Curriculum Learning & Hard Instance Mining.** Curriculum Learning (CL) (Bengio et al., 2009; Hacohen & Weinshall, 2019; Graves et al., 2017) sorts the data based on their hardness as measured by some scoring function ( ex., predictive entropy, softmax margin score, etc. ). It presents the data points during training in the order of increasing hardness. Similarly, Hard Instance Mining (HIM) (Zhou et al., 2020) reduces the weight of the easy example and increases the weight on hard inputs to promote hard-example learning. We point out that our method is only conceptually related to these works via input hardness. Our method helps the student with hard examples by providing explicitly discounted help $g$. We learn the helper function $g$ (through teacher representation) that decides whether the student needs help on a given input. Thus, we do not prioritize learning hard examples keeping in mind the fact that student capacity is much smaller than the teacher.

## 6 CONCLUSION

We develop a new knowledge distillation method that utilizes teacher predictions in novel ways combining predictive targets with scaffolding the student on hard-to-learn points by means of a guide function. Our method is particularly relevant when there is large gap between student and teacher capacities. We show that our method allows for convergence to better minima based on two key properties. Our guide function allows for censoring hard-to-learn examples, and the predictive targets set by the teacher on remaining points allow for eliminating bad local minima and smoothening the resulting student loss landscape. Against vanilla KD, we achieve improved performance and are comparable to more intrusive techniques that leverage feature matching on benchmark datasets.

### ACKNOWLEDGMENTS

This research was supported by Army Research Office Grant W911NF2110246, the National Science Foundation grants CCF-2007350 and CCF-1955981, AFRL Contract no. FA8650-22-C-1039, and a gift from ARM Corporation.

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

## A APPENDIX

### A.1 DETAILS FOR ILLUSTRATIVE EXAMPLE (1D INTERVALS)

**Dataset Overview.** We generate a synthetic toy dataset with one dimensional features $x \in [0, 9]$ and binary class labels $y \in \{\text{Red}, \text{Blue}\}$. We use $\sigma(x)$ to denote the sigmoid function with a scaled by parameter $\kappa > 0$, i.e., $\sigma(x) = \frac{1}{1+\exp(-\kappa x)}$.

**Function Classes.** Let $\mathcal{H}$ be the 1-interval function class parametrized by two variables $\{a, b\}$, i.e., for $h \in \mathcal{H}$

$$h(x; a, b) = \sigma(x - a) - \sigma(x - b); \;\; 0 < a < b < 9$$

Similarly, let $\mathcal{F}$ be the 2-interval family parametrized by four variables $\{a, b, c, d\}$, i.e., for $f \in \mathcal{F}$

$$f(x; a, b, c, d) = h(x; a, b) + h(x; c, d); \;\; 0 < a < b < c < d < 9$$

Note that any function in $\mathcal{H}$ behaves as an indicator for the interval $(a, b)$. Similarly, any function in $\mathcal{F}$ behaves as an indicator for two exclusive intervals $\{(a, b), (c, d)\}$.

**Data Generation.** We assume that the data is generated using the function $f^* \in \mathcal{F}$ with parameters $(a^*, b^*, c^*, d^*)$. Dataset is sampled with balanced data from both classes. We label $x$ as red if $f^*(x) < 0.5$, otherwise we label the point as blue. We sample 1000 i.i.d. data points as the training set and 100 data points as the test set. Figure 1 shows the train data. We draw an independent validation set of 100 data points for hyper-parameter tuning.

**Large Capacity Teacher** $T$ belongs to the 2-interval function class $\mathcal{F}$ and is learnt with all the training data points. We learn the teacher with cross-entropy loss. We use the SGD optimizer with momentum 0.9, learning rate 0.1, weight decay 0.01, and minimize the loss for 200 epochs. Note that the teacher recovers the underlying function $f^*$ as shown by the two intervals in Figure 1.

**Capacity Constrained Student** $S$ belongs to the 1-interval function class $\mathcal{H}$ and it has access to all the training dataset. Note that best possible hypothesis in $\mathcal{H}$ cannot recover the performance of the function $f^*$ and hence the student will have to settle on one of the many local minima. We show these minima as well the contour plot for the student in the Figure 1. We learn the student with three different loss functions ( cross-entropy $\mathcal{L}_{CE}$, vanilla KD $\mathcal{L}_{KD}^{\tau,\alpha}(\mathbf{s})$, and DiSK Algorithm 1 ). We use similar training setup as the teacher in terms of the optimizer and training steps. For DiSK method, our guide function $g$ has similar capacity as the student but utilizes the teacher features to learn the decision as to which points are hard-to-learn for the student. For both, KD and DiSK, we scan the $\alpha$ hyper-parameter over the range $\{0.0, 0.1, 0.5, 0.9, 1.0\}$. Similarly, we scan the temperature $\tau$ in the range $\{1, 2, 4\}$.

For DiSK, we scan the different hyper-parameters in the following ranges: (a) $\tau_s \in \{1, 2, 4\}$, (b) $K \in \{1, 2, 3\}$, (c) $\lambda_{\min} \in \{0.01, 0.1, 1, 5, 10\}$, (d) $\lambda_{\max} \in \{1, 5, 10, 20, 50, 100, 1000\}$, (e) Budget $\delta \in \{0.1, 0.05, 0.0\}$, and (f) $\lambda_T \in \{20, 50\}$. We replace the $\arg\min$ in Algorithm 1, with three gradient steps.

Note that although the hyper-parameter scan looks daunting, the default hyper-parameters: $\tau_s = \tau$ (teacher temperature), $K = 2$, $\lambda_{\min} = 0.1$, $\lambda_{\max} = 50$, $\delta = 0.0$ (approximate error of the global minima), $\lambda_T = 50$, work well in this setup as well as the 2D gaussian example described below.

**Vanilla KD suffers from local minima.** The loss landscape of the Vanilla KD contains many local minima (see Figure 1(b)). Since there is a big gap between student and teacher capacity, the teacher is unable to help the student discern between these bad minima. Hence, Vanilla KD leads to one of the bad local minima with high probability (see Table 1).

**DiSK** *removes bad local minima.* In contrast, DiSK deletes harder points from the landscape and as a result settles onto the global minima for $S$ with high probability (see Figure 1(c) and Table 1 where one cluster of blue points have been removed). Note that this also removes bad minima from the $S$ loss landscape. Finally, DiSK learns the student that has the best performance.

### A.2 DETAILS FOR ILLUSTRATIVE EXAMPLE (2D GAUSSIANS)

**Dataset Overview & Data Generation.** We generate a synthetic toy dataset with two dimensional features $\mathbf{x} \in \mathbb{R}^2$ and three class labels $y \in \{\text{Red}, \text{Green}, \text{Blue}\}$. We generate six cluster centers. We

assign a color to each cluster center and spread input features around these centers. Below we list the cluster centers along with their class labels.

- $(0, 0)$, Red
- $(1.5, 0)$, Blue
- $(3, 0)$, Green
- $(0, 1.5)$, Blue
- $(1.5, 1.5)$, Green
- $(3, 1.5)$, Red

Given a cluster center $\mathbf{c}$ , we draw input features $\mathbf{x}$ using a Gaussian ball with radius $r = 0.05$ around the center using multi-variate Gaussian $\mathcal{N}(\mathbf{c}, r\mathbb{I})$, where $\mathbb{I}$ is the Identity matrix.

Figure 2a shows the labelled data. We sample 1000 i.i.d. data points as the training set and 1000 data points as the test set with equal representation from all three classes.

**Function Classes.** We use two feed-forward neural networks as function classes in this example. Let $\phi(\cdot)$ denote the Batch-Norm followed by ReLU operation.

Let $\mathcal{H}$ be the two feed-forward layer neural network. Any $h \in \mathcal{H}$ can be written as

$$h(x) = W_2\phi(W_1x)$$

where $W_1 \in \mathbb{R}^{2\times2}$ and $W_2 \in \mathbb{R}^{3\times2}$. Note that $h$ has only two neurons and hence a very small network.

Let $\mathcal{F}$ be the three feed-forward layer neural network. Any $f \in \mathcal{F}$ can be written as

$$f(x) = \hat{W}_3\phi(\hat{W}_2\phi(\hat{W}_1x))$$

where $\hat{W}_1 \in \mathbb{R}^{8\times2}$, $\hat{W}_2 \in \mathbb{R}^{16\times8}$ and $\hat{W}_3 \in \mathbb{R}^{3\times2}$.

Note that $f$ has 8 neurons in first and 16 neurons in the second layer. The final layer in above networks is the classifier layer that transforms the features into the class probabilities.

**Large Capacity Teacher** $T$ is a 3 layer neural network with 8, 16 and 3 neurons. In between each feed-forward layer, we have batch-norm and ReLU activation non-linearity. We point out that the teacher being an over-parameterized network in this feature space, easily learns the correct decision boundary. We show this decision boundary in the Figure 2a. We learn the teacher with cross-entropy loss. We use the SGD optimizer with momentum 0.9, learning rate 0.1, weight decay 0.01, and minimize the loss for 200 epochs.

**Capacity Constrained Student** $S$ is a 2 layer neural network with 2 and 3 neurons. Similar to the teacher, we have batch-norm and ReLU non-linearity in between the feed-forward layers. Since the student is severely constrained as compared to the teacher, it suffers in learning the task. Different training runs lead to some popular local minima. We show the teacher solution as well as the student local minima in Figure 2a. For DiSK method, our guide function $g$ has similar capacity as the student but utilizes the teacher features to learn the decision as to which points are hard-to-learn for the student. The contour plots for the student models under KD loss and DiSK loss are shown in Figure 2b-2c using the visualization toolkit described in Li et al. (2018). We following similar setup for hyper-parameter tuning as in Sec. A.1.

We see a similar result as in 1D example. KD suffers from bad local minima and converges to the global minima with only 43% of the initializations. Differently, DiSK escapes the local minima solutions and focus on the learnable part of the input space as shown in Figure 2c. Our method converges to the global minima with very high probability ( see Table 1).

### A.3 DATASET DETAILS

We use publicly available CIFAR-100 (Krizhevsky, 2009), Tiny-Imagenet (Le & Yang, 2015) and ImageNet-1K (Russakovsky et al., 2015) datasets. CIFAR-100 contains 50K training and 10K test images from 100 classes with size $32 \times 32 \times 3$. While Tiny-Imagenet contains 100K training and

10K test images from 200 classes with size $64 \times 64 \times 3$. Imagenet contains 1.2M training and 100K test images from 1000 classes with size $224 \times 224 \times 3$.

For the CIFAR-100 and Tiny-Imagenet datasets, we use the standard data augmentations including 'RandomCrop', 'RandomHorizontalFlip', 'AutoAugment'(Cubuk et al., 2019), 'Cutout' (DeVries & Taylor, 2017), and 'Mean-Std-Normalization'. We use the same augmentation strategy in all our experiments, across baselines and different model families.

For the Imagenet-1K dataset, following the previous work Chen et al. (2022), we use the 'Random-Crop', 'RandomHorizontalFlip', and 'Mean-Std-Normalization'.

### A.4 MODEL DETAILS

In this section, we list the model characteristics as well as their accuracy obtained using standard cross-entropy (CE) loss. Table 5 lists all the models used in large capacity mismatch setting. While Table 6 lists all the models in the small capacity mismatch setting. Below, we describe individual model for completeness.

**Large Student-Teacher Capacity Mismatch** All models in the Table 5 belong to the same ResNet family and use the standard 'BasicBlock' as the building block. It consists of a convolutional block, followed by four residual block stages, followed by the adaptive average pooling layer and the classifier layer. Different capacity models in this family differ only in the number of repetitions of the residual block and the number of filters in each stage. Below, we write the different of repetitions and the number of filters for the four different residual stages.

- *ResNet34* has $[64, 128, 256, 512]$ filters and repeats the 'BasicBlock' $[3, 4, 6, 3]$ times.
- *ResNet18* has $[64, 128, 256, 512]$ filters and repeats the 'BasicBlock' $[2, 2, 2, 2]$ times.
- *ResNet10* has $[64, 128, 256, 512]$ filters and repeats the 'BasicBlock' $[1, 1, 1, 1]$ times.
- *ResNet10-$\ell$* has $[32, 64, 128, 256]$ filters and repeats the 'BasicBlock' $[1, 1, 1, 1]$ times.
- *ResNet10-m* has $[16, 32, 64, 128]$ filters and repeats the 'BasicBlock' $[1, 1, 1, 1]$ times.
- *ResNet10-s* has $[8, 16, 32, 64]$ filters and repeats the 'BasicBlock' $[1, 1, 1, 1]$ times.
- *ResNet10-xs* has $[8, 16, 16, 32]$ filters and repeats the 'BasicBlock' $[1, 1, 1, 1]$ times.
- *ResNet10-xxs* has $[8, 8, 16, 16]$ filters and repeats the 'BasicBlock' $[1, 1, 1, 1]$ times.

**Small Student-Teacher Capacity Mismatch** Definitions of all models in the Table 6 are borrowed from Chen et al. (2022). We refer the reader to their official github repository (`https://github.com/DefangChen/SimKD.git`) for the exact definition. We trained these models on our end using the data augmentations mentioned above and found that our cross-entropy baseline as well as the vanilla KD baselines are much better than the ones reported in their work.

**Guide Function** Our guide function $g$ is a three layer feed-forward network. It uses the last layer features and logits of the teacher as the input. It has 64, 128, and 1 neurons in the three layers. We include batch-norm followed by ReLU non-linearity between these layers. The final layer contains a sigmoid activation to contain the scaler output in the range $[0, 1]$.

**Warm Start** We note that we warm start each student model by first training them with cross entropy loss without teacher. We observe that the warm start benefits both DiSK and KD. Note that, we do not change the algorithms. We only start from a CE pre-trained student model.

**Table 5:** Models used in large capacity mismatch setting along with storage and computational requirements.

| Architecture | | CIFAR-100 | | | Tiny-Imagenet | | |
|---|---|---|---|---|---|---|---|
| | | CE Acc. | MACs | Params | CE Acc. | MACs | Params |
| **Teacher** | ResNet10-$\ell$ | 71.99 | 64M | 1.25M | 52.14 | 255M | 1.28M |
| | ResNet10 | 75.25 | 253M | 4.92M | 56.04 | 1013M | 5M |
| | ResNet18 | 76.56 | 555M | 11.22M | 62.48 | 2221M | 11.27M |
| | ResNet34 | 80.46 | 1159M | 21.32M | 63.06 | 4637M | 21.38M |
| **Student** | ResNet10-xxs | 32.05 | 2M | 13K | 17.44 | 8M | 15K |
| | ResNet10-xs | 42.99 | 3M | 28K | 25.89 | 12M | 31K |
| | ResNet10-s | 52.16 | 4M | 84K | 34.65 | 16M | 90K |
| | ResNet10-m | 65.24 | 16M | 320K | 44.74 | 64M | 333K |

**Table 6:** Models used in in small capacity mismatch setting along with storage and computational requirements.

| Architecture | | CIFAR-100 | | |
|---|---|---|---|---|
| | | CE Acc. | MACs | Params |
| Teacher | ResNet32x4 | 81.45 | 1083M | 7.4M |
| | Wide-ResNet-40-2 | 78.41 | 327M | 2.25M |
| Student | ResNet8x4 | 73.89 | 177M | 1.2M |
| | ShuffleNetV2 | 73.74 | 44.5M | 1.4M |
| | Wide-ResNet-16-2 | 74.29 | 101M | 700K |
| | Wide-ResNet-40-1 | 72.81 | 83M | 570K |
| | MobileNetV2x2 | 69.24 | 22M | 2.4M |

**Table 7:** Imagenet-1K: We pick some student and teacher configurations to show that we can scale DiSK to ImageNet dataset with significant improvements in Top-1 accuracy. We borrow model definitions from timmWightman (2019) repository including the convolutional and transformer vision models.

| Teacher | Student | CE | KD | DiSK |
|---|---|---|---|---|
| ResNet50 | ResNet18 | 69.73 | 71.29 | 72.35 |
| ViT-Large (Patch 16, Res. 224) | ViT-Tiny (Patch 16, Res. 224) | 75.45 | - | 77.86 |

## A.5 HYPER-PARAMETERS

For both, KD and DiSK, we scan the $\alpha$ hyper-parameter over the range $\{0.0, 0.1, 0.5, 0.9, 1.0\}$. As per recommendations from previous works(Chen et al., 2022; Cho & Hariharan, 2019; Tung & Mori, 2019), we use $\tau = 4$ as the temperature in Eq. 1.

For DiSK, we scan the different hyper-parameters in the following ranges: (a) $\tau_s \in \{1, 2, 4\}$, (b) $K \in \{1, 3, 5, 10, 20, 50\}$, (c) $\lambda_{\min} \in \{0.01, 0.1, 1, 5, 10\}$, (d) $\lambda_{\max} \in \{1, 5, 10, 20, 50, 100, 1000\}$, (e) Budget $\delta$ within $0.2$ distance from the cross-entropy trained student model's error, and (f) $\lambda_T \in \{20, 50\}$. We replace the $\arg\min$ in the Algorithm 1, with three SGD steps over the entire dataset. For all our experiments (both KD and DiSK), we use the popular cosine learning rate scheduler for the SGD optimizer $0.1$ learning rate, $0.9$ momentum and $5e - 4$ weight decay. We use $200$ as the batch size.

Note that the default hyper-parameters: $\tau_s = \tau$ (teacher temperature), $K = 20$, $\lambda_{\min} = 0.1$, $\lambda_{\max} = 50$, $\delta$ = approximate error of the global minima (replaced by the cross-entropy error), $\lambda_T = 50$, work well in most of our experiments.

We point out that the denominator $N$ in the budget constraint should be calibrated for the correct numerical implementation. Instead of $N$ we use the number of wrong student predictions as the normalizer, i.e., $\sum_{i=1}^{N} y_i \neq \arg\max_y s_y(x_i)$ as this is the term that appears in the Eq. 2 alongside the $g$ term in the budget constraint.

We note that we use similar parameters to train with CE in ResNet based models. Improving CE training would improve DiSK and KD as well since both are initialized with the CE trained model.

## A.6 DIFFERENT EXPERIMENT SETUPS

**Scaling upto ImageNet setting.** We show that DiSK scales easily to the large-scale ImageNet-1K dataset. We train two configurations with DiSK, namely (a) ResNet18 student and ResNet50 teacher, and (b) ViT-Tiny student and ViT-Large teacher. We borrow these models from the timm(Wightman, 2019) library. Table 7 shows the DiSK performance on these two configurations along with the baseline. It clearly shows that DiSK scales well to ImageNet-1K setup and achieves significant improvements over the baselines.

## A.7 RELATED WORK CONTINUED

We continue a section of related work here due to the page constraint in the main part.

**Privileged Information.** Vapnik & Izmailov (2015) propose the 'learning under privileged informa-tion' (LUPI) framework wherein a support vector machine is trained using privileged information

**Table 8:** DiSK performance against feature matching KD on CIFAR-100: Similar setup as in Table 4. We integrate DiSK within SimKD (Chen et al., 2022) (see Appx. A.8) The gains of using DiSK over KD and using SimKD + DiSK over SimKD are reported. Feature matching KD baselines are due to Chen et al. (2022).

| Architecture | | Response Matching KD | | | | | Feature Matching KD | | | | |
|---|---|---|---|---|---|---|---|---|---|---|---|
| | | Accuracy (%) | | | | | Accuracy (%) | | | | |
| Teacher | Student | Teacher | CE | KD | DiSK | Gain | FitNet | SemCKD | SimKD | SimKD + DiSK | Gain |
| Wide-ResNet-40-2 | ResNet8x4 | 78.41 | 73.89 | 75.15 | **76.05** | **0.9** | 75.02 | 75.85 | 76.75 | **77.13** | **0.38** |
| | Wide-ResNet-40-1 | | 72.81 | 74.44 | **75.92** | **1.48** | 74.17 | 74.4 | 75.56 | **76.21** | **0.65** |

unavailable during the inference stage. Later, Lopez-Paz et al. (2016) unify LUPI and vanilla KD into generalized distillation, wherein the teacher is learned using the privileged information. Next, the student is trained using the ground truth and the teacher labels. These works rely on privileged information in the application domain and are shown to work on toy setups. Since our guide function, $g$ is available only during training, it can be thought of as privileged information from a teacher.

## A.8  EXTENSIONS BEYOND VANILLA KD.

In this section, we discuss potential extension of our method beyond Vanilla KD to feature based distillation as well as self-supervised distillation.

*Feature based KD methods.* Let $f_s$ and $f_t$ denote the features for the student and teacher respectively and $\Psi$ denote the operator such that $\Psi(f_s)$ lies in the same feature space as $f_t$ . Commonly used feature transfer strategy is to minimize the distance between these two representations via loss function such as mean-squared error as shown below by the loss $\mathcal{L}_{ft}$.

$$\mathcal{L}_{ft} = -\frac{1}{N}\sum_{i=1}^{N}\|\Psi(f_s(\mathbf{x}_i))-f_t(\mathbf{x}_i)\|^2; \quad \mathcal{L}_{ft-g} = -\frac{1}{N}\sum_{i=1}^{N}(1-g(\mathbf{x}_i))\|\Psi(f_s(\mathbf{x}_i))-f_t(\mathbf{x}_i)\|^2 \quad (6)$$

A simple extension of this feature alignment loss to the selective distillation is shown by the loss $\mathcal{L}_{ft-g}$ that weighs each data point with the helper function decision. Table 8 shows this feature matching extension for the SimKD(Chen et al., 2022) scheme. We leave question of finding better selective distillation losses in feature alignment to future work.

*Self-Supervision.* We can substitute the teacher with a moving average of the student to help guide in the distillation process (a surrogate that helps in learning the hints).

*Other Domains.* Although our work focuses mainly on models in the vision domain. We point out that conceptually the function $g$ is applicable on models in other domains albeit with some modifications. For instance, we can utilize the censoring mechanism in sequential decision making with recurrent neural networks (Kag et al., 2020; Kag & Saligrama, 2021). We already show that DiSK can be applied to the vision transformers in the Table 7. We leave the application and modifications to DiSK for transformers in language domain for future research.

