# OpenReview forum: "Scaffolding a Student to Instill Knowledge"
_ICLR.cc/2023/Conference — ICLR 2023 poster_

### Official Review · Reviewer_DdZ6 · 2022-10-25

**Confidence:** 4
**Correctness:** 4
**Technical Novelty And Significance:** 3
**Empirical Novelty And Significance:** 3
**Recommendation:** 6

**Clarity, Quality, Novelty And Reproducibility:**

**Clarity:**
- As mentioned above, the paper should be restructured so that the intro gives the high level idea, then the method is explained, then the theoretical arguments and toy experiments are presented. Even having the "if g(x) $\approx$ 1, teacher discounts the input" is a level of technical detail not appropriate for the intro.
- Egregious abuse of negative vspace made it hard for me to parse page 2... I couldn't find the continuation of the previous page because it appeared to be part of the figure caption.
- Equation 2 is nicely presented and clearly broken down into its components.
- p. 5 "contraints the helper g" -> constrains
- p. 6 "the indices guide function helps are picked as" - ??
- Table 4 is awkwardly located in the related work section, after the results are already written up.
- p. 7 "where student is very less-capacited compared to teacher model" -> both spelling and grammar errors. Where the student has much less capacity
- incomplete sentence on p. 7 under the bolded heading "Experiment results."
- p. 8 "as to remedy to"

**Quality:**
- As mentioned above, both the toy experiments and the scaled up experiments are compelling evidence for the usefulness of the method.
- The results could be presented more carefully so as not to be misleading. Why does Table 3 compute gain as the gain over CE rather than KD, since KD is the more relevant baseline? More importantly, the convention is to bold the highest scoring method. In Table 4, SimKD scores highest for Resnet8x4 (rows 1 and 5), so it should be bolded.
- The relevance of the feature matching baselines in table 4 is not well explained.
- The point about reaching near teacher accuracy with a model with many fewer parameters is a good one, and could be emphasized more. This could be impactful for a number of applications.

**Originality:**
- To the best of my knowledge, this work is original in proposing that the teacher not penalize the student for learning examples that are too difficult in order to preserve model capacity in the student.

**Strength And Weaknesses:**

A strength of the paper is in the quality of the experiments. Starting with the toy experiments, the authors clearly demonstrate how their approach is valuable in helping a low capacity student find the best minimum point in the loss optimization landscape; Figure 1 convincingly shows how the gated teacher loss function reduces local minima. Beyond the toy experiments, the paper tests on real datasets with standard large resnet models, and shows consistent improvements over prior work.

A weakness of the paper is its organization. The intro extends for almost 4 pages, and includes detailed experiments and proofs that are wholly inappropriate for the intro. The method has not been fully explained, and yet the paper is diving into experiments and theoretical arguments to show that it works.

**Summary Of The Paper:**

The paper proposes a method for improving knowledge distillation from a high capacity teacher model to a low capacity student model. In addition to training the student to match the teacher's outputs with cross-entropy, the paper proposes learning a 'guide' (or essentially a gating) function from the teacher's output that reduces the weight placed on difficult examples when training the student. Experiments reveal that this smooths the optimization landscape for the student, enabling it to find the best minimum even when it does not have the capacity to represent the ground truth function. In addition to results on toy environments, the experiments examine CIFAR-100 datasets with standard architectures, and show an improvement over vanilla KD and other KD techniques.

**Summary Of The Review:**

Overall, the paper is poorly organized but the results are compelling. I think it will be of interest to the community.

---

> ### Author Response · Authors · 2022-11-14
> **Introduction, re-organization and Table 4**
>
> **Long Introduction. Re-organization.**
>
> Our intention in having toy examples precede the exact algorithmic specification is to present the conceptual aspects of the method, and illustrate the key favourable structural properties (censoring, eliminating minima, and better generalisation) that the scaffolding approach introduces. The method of section 3 is ultimately only one instantiation of the conceptual approach being proposed. We believe that this is a clean way of ensuring that readers appreciate the key ideas, and as mentioned by the other reviewers, it is effective in doing so.
>
> Nevertheless, we agree that the introduction is long, and that some readers would prefer to see the precise method first. To address this, we have edited to a) discuss the contributions right after the discussion of structural properties (censoring, etc.) in the introduction, b) introduce a section break to separate out the illustrative examples, and c) in the beginning of the new section 2, explicitly include a link to the precise method description (new section 3) in order to accommodate readers that would prefer to go through this before any experiments. We hope that these constitute an acceptable compromise.
>
> **Reporting gain as the gain over CE rather than KD.  Bold the highest scoring.**
>
> *Gain*. We agree with your observation regarding the gains over KD, and have edited accordingly in the new revision.
>
> *Bolding the highest scoring method*. SimKD introduces significant changes in student architecture and in particular increases student capacity. Hence, we think that it is not directly comparable to our method - the point of comparing to it is mainly completeness (see below). This lack of comparability is the reason we did not emphasize the SimKD accuracies in Table 4, since we believe that this distracts from the main point of comparison (vanilla KD). We have included a note on the table to mention this in the revision.
>
> **The relevance of the feature matching baselines in Table 4**
>
> Feature matching KD methods attempt to align internal feature representations of students with the teacher rather than only the logits. Typically, this results in an increased training cost, but an improved backbone that with recent methods like SimKD improves upon student accuracy, making them some of the strongest KD baselines in the literature. The purpose of including these feature matching baselines in Table 4 is thus completeness with respect to KD approaches, and also to express the surprising observation that DiSK remains competitive with, and often outperforms, even such strong methods in the large student capacity regime. We have included a brief discussion of this aspect (last paragraph before Experiment Results in pg. 7) in the revision.
>
> **Emphasizing more about reaching near teacher accuracy with a model with many fewer parameters**
>
> We updated our contribution list (pg. 2) to emphasize more on this aspect with specific reference to the experiments section.

---

> > ### Comment · Reviewer_DdZ6 · 2022-11-21
> > **Thank you for your response**
> >
> > Thank you for your detailed response and for incorporating some of my suggestions.

---

### Official Review · Reviewer_XDpR · 2022-10-25

**Confidence:** 4
**Correctness:** 3
**Technical Novelty And Significance:** 3
**Empirical Novelty And Significance:** 3
**Recommendation:** 8

**Clarity, Quality, Novelty And Reproducibility:**

The paper is well written with clarity. And the idea proposed here is novel.

Minor edits:
- two "bound" in page 2 under section "Good Generalization".

**Strength And Weaknesses:**

Strength
- The paper is well motivated. I especially like the introduction section (despite being long) which clearly conveys the important ideas with two illustrating examples.
- The methodology is well formulated.
- Experiments are comprehensive with detailed discussions.

Weaknesses
- The selection of budget is an important factor in the whole algorithm, but there seems not to be discussions on how users should choose in practice.
- The stability and efficiency of the proposed algorithms are not discussed.



**Summary Of The Paper:**

This paper introduces a novel knowledge distillation (KD) method aims at addressing the capacity mismatch between teacher and student models. The authors propose to additionally learn a weight function to mask hard to learn examples, and thus increase the performance of student models.

**Summary Of The Review:**

This paper provides a novel approach for knowledge distillation by distinguish hard to learn samples for students, and proves to provide a student model with smoother landscape, fewer local minima, and thus better generalization errors. I think the idea proposed is novel, and the paper is well motivated with very intuitive examples to present the ideas. It is a good paper for ICLR.

---

> ### Author Response · Authors · 2022-11-14
> **Budget constraint selection, stability and efficiency**
>
> **The selection of budget is an important factor in the whole algorithm, but there seems not to be discussions on how users should choose in practice.**
>
> Note that the tested hyper-parameters were reported in detail in Appendix A.5.
>
> *Motivating the choice of $\delta$*: Viewing the scaffold as a way to allow the student to interpolate the training data suggests that the budget needed for $g$ is roughly equal to the error of cross-entropy trained model without teacher supervision (so that in the worst case the log-penalised scaffold can eliminate these errors). With this intuition, we initialize $\delta$ around this CE error rate and tune it over a small interval around this value of width $0.2$. If $\delta$ is much smaller than this initial value, then the guide is very limited, and student training does not get any help from scaffolding. On the other hand, if the budget is set to a much larger value, then guide dominates and student does not receive any signals from the teacher. We included this discussion in Definitions and Formulations Section (starting 'We view the scaffold' after Budget Constraint Relaxation paragraph, pg. 5) in the revision.
>
> **The stability and efficiency of the proposed algorithms are not discussed.**
>
> *Stability:* We have reported the variance of our method in Experiments (Methods paragraph in 'as the variance of 3 trials in our experiments is not larger than 0.1 in most of the cases'). Hence, DiSK is stable to initialization and randomness of batches.
>
> *Computational efficiency:* Different from KD, DiSK trains an extra guide function. The guide function is a three layer neural network (Function $g$ construction, pg. 5). Compared to the student network which is a usually a variant of ResNet, training cost of guide function is very small. We use standard gradient descent method to train both networks. Hence, the training costs of DiSK are comparable to standard KD. We included this discussion in Definitions and Formulations Section (last paragraph of the section in pg. 6) in the revision.

---

### Official Review · Reviewer_sPvE · 2022-10-25

**Confidence:** 2
**Correctness:** 3
**Technical Novelty And Significance:** 3
**Empirical Novelty And Significance:** 3
**Recommendation:** 6

**Clarity, Quality, Novelty And Reproducibility:**

The paper is well written and provides a novel approach to performing distillation when the student model is significantly less resourced than the teacher model.

**Strength And Weaknesses:**

The greatest strength of the work is the considerable gain in accuracy compared to cross-entropy or kullback-leibler. It is very impressive for such resource constrained models. Figure 1 and the example provided is excellent to provide an overview of what can be achieved in regards to simplifying the loss landscape.

One of the weakness in the paper is that it is unclear what the guide function model is. In one part of the paper it is described as "obtained by using an MLP on softmax outputs of the teacher" in another place it is described as taking the teacher's last layer features and the prediction probabilities as input.

Another weakness is that decisions are presented without an explanation. Why relax the guide function to be continuous? Why is the budge constraint necessary?



**Summary Of The Paper:**

The authors propose an approach to distillation where the teacher provides both a predictive target and scaffolds the student's prediction by censoring hard-to-learn examples. In the case where the student has far fewer parameters than the teacher this scaffolding leads to a smoother loss landscape for the student and thus the student encounters fewer local minima. The authors call their approach DiSK (distilling selective knowledge).

**Summary Of The Review:**

The authors provide a novel approach to performing distillation in which scaffolding allows hard to learn examples to not impact the student. The authors note that an implementation would be provided in the final version, and as a reader I desperately wanted it. I believe it would clarify some questions i have in regards to the guide functions implementation.

---

> ### Author Response · Authors · 2022-11-14
> **Guide function construction and Budget constraint**
>
> **How is $g$ constructed?**
>
> We released the code (see shared link above).
>
> For reference we provide additional details below.
> One of the major resources supplied by a well trained teacher model is the rich featurization it develops - this is critical in drawing any signal from the data. For this reason, as described in 'Function $g$ Construction' (pg. 5), in our experiments on deep neural nets, the guide function is a 3 layer feed-forward network. Its inputs are 1) The last layer features of the teacher, and 2) the logits of the teacher output. The number of hidden neurons are 64 and 128, and the output layer has one neuron that takes a sigmoid. We improved the clarity in all occurrences in the paper to avoid confusion. Note that due to the simplicity of the motivating examples (1D intervals and 2D Gaussians), the guide functions for these examples are smaller and reported separately in the appendix.
>
> **Why relax the guide function to be continuous? Why is the budget constraint necessary?**
>
> Roughly speaking, the guide function is trying to classify query points as "easy" or "hard" for the student, based on signals from the data and the teacher model. This consideration motivates both of these choices:
>
> *Smoothening for soft classification:* smoothening the guide to [0,1] results in a soft classification task. This is the standard methodology used in classification - indeed, the idea is to slowly evolve from non-predictive estimates towards the edges of the interval [0,1] as training proceeds. Taking this soft approach then gives us a convenient way to use standard learning methods, in particular gradient based learning, without having to explicitly deal with a thresholding.
>
> *Budget constraints to prevent triviality:* Note that we do not have explicit labels for the guide function telling us which points are easy or hard - we need to discover this from the data and the teacher's behaviour. In trying to distinguish the points as easy or hard, it is possible that an unconstrained guide function says that every point is hard - this would actually minimise our distillation term (4), but is a useless solution because this means that the student function gets no training signal from the teacher on such "hard" points. As observed in the introduction (last paragraph, starting However, we must...), we need to counteract this tendency of the guide function to ensure that the student learns. The budget constraint is a simple way to do this. Effectively, it demands that not too many points are declared to be hard, so that the teacher signals are exploited by the student. Of course, exactly what "too many points" means is indeterminate, and we resort to setting this using a hyper-parameter scan, represented as $\delta$ in our loss (5).

---

### Decision · Program_Chairs · 2023-01-20

**Decision:**

Accept: poster

**Justification For Why Not Higher Score:**

Lack of standard ImageNet experiments raises uncertainty regarding the potential impact of this paper.


**Justification For Why Not Lower Score:**

N/A

**Metareview: Summary, Strengths And Weaknesses:**

This paper presents a novel approach to distillation called DiSK (distilling selective knowledge) where the teacher provides both a predictive target and censors hard-to-learn examples for the student. This scaffolding leads to a smoother loss landscape for the student, resulting in fewer local minima. The authors provide synthetic examples to support their method and present a bound for the performance of the student model.

Overall, the idea is well motivated and the synthetic examples are effective in illustrating the technique and the potential benefits of this approach. The bound is also a nice contribution. It would be interesting to explore driving the censoring based on the student's model or both the student and teacher. A weakness of this paper is the lack of standard ImageNet experiments, as techniques that perform well on Cifar and tiny ImageNet often do not generalize to standard ImageNet and other more challenging tasks. This cast some doubt on the method's potential impact.
I encourage the authors to address any unresolved issues raised by the reviewers.


**Note From Pc:**

if the above contains the word "oral" or "spotlight" please see: "oral" presentation means -> notable-top-5% and "spotlight" means -> notable-top-25%. As stated in our emails, we are disassociating presentation type from AC recommendations